# FREEZE AND CLUSTER: A SIMPLE BASELINE FOR REHEARSAL-FREE CONTINUAL CATEGORY DISCOVERY

## ABSTRACT

This paper addresses the problem of Rehearsal-Free Continual Category Discovery (RF-CCD), which focuses on continuously identifying novel class by leveraging knowledge from labeled data. Existing methods typically train from scratch, overlooking the potential of base models, and often resort to data storage to prevent forgetting. Moreover, because RF-CCD encompasses both continual learning and novel class discovery, previous approaches have struggled to effectively integrate advanced techniques from these fields, resulting in less convincing comparisons and failing to reveal the unique challenges posed by RF-CCD. To address these challenges, we lead the way in integrating advancements from both domains and conducting extensive experiments and analyses. Our findings demonstrate that this integration can achieve state-of-the-art results, leading to the conclusion that *"in the presence of pre-trained models, the representation does not improve and may even degrade with the introduction of unlabeled data."* To mitigate representation degradation, we propose a straightforward yet highly effective baseline method. This method first utilizes prior knowledge of known categories to estimate the number of novel classes. It then acquires representations using a model specifically trained on the base classes, generates high-quality pseudo-labels through k-means clustering, and trains only the classifier layer. We validate our conclusions and methods by conducting extensive experiments across multiple benchmarks, including the Stanford Cars, CUB, iNat, and Tiny-ImageNet datasets. The results clearly illustrate our findings, demonstrate the effectiveness of our baseline, and pave the way for future advancements in RF-CCD.

## 1 INTRODUCTION

Humans possess the ability to continuously learn new knowledge in ever-changing environments with limited supervision. Inspired by this capability, several studies have proposed the problem of continual novel class discovery (Roy et al., 2022; Joseph et al., 2022), aiming to enable models to continuously capture new categories from unlabeled data. Such a continuous learning strategy can be applied to a variety of artificial agents, for instance, allowing robots to autonomously learn in new environments (Dai et al., 2024; Kejriwal et al., 2024). However, this is a highly challenging problem, as it requires models to have the plasticity to discover new classes while avoiding catastrophic forgetting with little supervision.

To address this problem, existing methods (Zhang et al., 2022b; Zhao & Mac Aodha, 2023; Joseph et al., 2022; Roy et al., 2022) often draw on learning techniques from the field of novel class discovery (Han et al., 2019; Gu et al., 2023; Zhang et al., 2023a; Fini et al., 2021), such as self-labeling (Fini et al., 2021) or pair-wise learning (Han et al., 2019), to discover novel classes and employ memory replay or generative feature-replay to prevent catastrophic forgetting in feature extractors and classifiers. However, these methods typically train from scratch, overlooking the development of foundational models and heavily relying on memory to store raw data, which can be impractical in privacy-sensitive and/or low-resource scenarios. Subsequently, Liu et al. (2023) propose a rehearsal-free baseline based on frozen pre-trained models, but provide limited insights into the use of frozen pre-trained models. Moreover, they compare their approach with earlier methods (Kirkpatrick et al., 2017; Li & Hoiem, 2017; Buzzega et al., 2020), while overlooking

recent advancements in continual learning, such as (Zhang et al., 2023b; Smith et al., 2023; Wang et al., 2022b). This oversight results in relatively limited experimental comparisons and as such the conclusions remain open for the task of continual class discovery. More importantly, they fail to address a crucial question: *beyond the challenges of continual learning and novel class discovery, what unique challenges does rehearsal-free continual category discovery (RF-CCD) face?*

To overcome the above limitations, we initially combine existing methods from two different fields and conduct extensive experiments on RF-CCD problem. Specifically, we select LwF (Li & Hoiem, 2017), CoDA-Prompt (Smith et al., 2023), and SLCA (Zhang et al., 2023b) for our analysis. To enable these continual learning methods to discover novel classes, we replace supervised losses with unsupervised ones, including Self-Labeling (Fini et al., 2021), PairWise (Han et al., 2021; Cao et al., 2022), and Self-Distillation (Wen et al., 2022) loss, summarized from the field of category discovery. Then, we rigorously test them across multiple benchmark datasets and probe the representation quality. Our experiments reveal that SLCA with self-distillation loss outperforms current methods (Liu et al., 2023; Wu et al., 2023; Roy et al., 2022). More importantly, we empirically found that *with the best combination learning strategy, continuous novel class discovery does not enhance, and can even degrade, the representational capacity of the model.* This is in stark contrast to supervised continual learning, which continuously improves the model's representational capabilities, highlighting a unique challenge for RF-CCD.

Based on our experimental observations, we propose a simple yet effective baseline method named "Freeze and Cluster" (FAC) to tackle the RF-CCD problem. Specifically, during the initial known-class learning stage, we fine-tune the representation using known classes, which is essential for adapting to downstream tasks. Concurrently, we perform over-clustering and progressively merge clusters until they align with the ground truth, thereby deriving the minimal distance between clusters. For subsequent novel class learning, we estimate the number of novel classes by over-clustering the data and iteratively merging clusters until the minimal distance is achieved. The remaining clusters represent the estimated number of novel classes. To discover these novel classes, we freeze the representation space and apply k-means clustering to group the novel classes, assigning pseudo-labels to each unlabeled data point. We then calculate the mean and variance for each identified cluster. Finally, classifiers are trained by sampling data points from each cluster based on their means and variances. In summary, FAC addresses the challenging issue of representation degradation by freezing the model's backbone in the novel class discovery stage.

To illustrate the unique challenges of RF-CCD and demonstrate the effectiveness of our proposed baseline, FAC, we conduct comprehensive experimental analyses on CUB, StanfordCars, TinyImageNet, and the challenging iNat2021 datasets. In summary, our contributions are three-fold:

- We conduct comprehensive experiments to illustrate that: 1) combining continual learning with novel class discovery methods can significantly surpass existing RF-CCD approaches; and 2) the best combination learning strategies do not improve, and can even degrade, the model's representational ability in RF-CCD.

- We propose a simple yet effective baseline, Freeze and Cluster, to address RF-CCD, which estimates the number of novel classes and discovers novel classes by learning classifier.

- We conduct experiments on CUB200, Scars196, Tiny-ImageNet, and iNat500, and our proposed baseline achieves state-of-the-art performance on these benchmarks compared to current continual learning methods, paving the way for subsequent developments.

## 2 RELATED WORK

### 2.1 CONTINUAL LEARNING

The goal of Continual Learning (CL) is to train a model to sequentially perform a series of tasks while only accessing the data of the current task and evaluating the model's performance on all tasks encountered so far. Continual learning methods aim to mitigate the catastrophic forgetting of previous task knowledge while enabling the model to flexibly learn new tasks. Existing continual learning work primarily focuses on sequential training of deep neural networks from scratch. Representative strategies include regularization-based methods such as LwF (Li & Hoiem, 2017) and Afec (Wang et al., 2021b), which retain the old model and selectively update parameters; replay-based methods

such as Gdumb (Prabhu et al., 2020), TMNs (Wang et al., 2021a), and DER (Buzzega et al., 2020), which approximate and restore previously learned data distributions in each new task; and architecture-based methods such as Coscl (Wang et al., 2022a), HAT (Serra et al., 2018), and DER (Yan et al., 2021), which allocate dedicated parameter subspaces for each incremental task.

**Continual Learning on Pretrained Model** Witnessing the significant improvement brought by powerful pre-training for downstream tasks, some recent methods have focused on exploring continual learning methods in the context of pre-trained models. SAM (Mehta et al., 2021) demonstrated the benefit of supervised pre-training for downstream continual learning tasks; L2P (Wang et al., 2022c) proposed updating the network with a small number of learnable parameters (prompts), and DualPrompt (Wang et al., 2022b) and CODA-prompt (Smith et al., 2023) further improved prompt learning methods and enhanced the model's continual learning capabilities. SLCA (Zhang et al., 2023b) studied the updating paradigm of pre-trained models and significantly improved prediction accuracy by lowering the learning rate of the backbone network. Meanwhile, some works mainly explored the learning of classifiers (Janson et al., 2022; Goswami et al., 2024; Panos et al., 2023; McDonnell et al., 2024). However, while those techniques are effective in supervised scenarios, their ability to address open-world problems remains to be explored.

## 2.2 CATEGORY DISCOVERY

**Novel Class Discovery** Novel Class Discovery (NCD) involves using the knowledge obtained from a labeled base dataset to learn and discover new classes in an unlabeled dataset. Existing methods in this field can be categorized into three groups based on the loss function used for clustering novel classes. 1) Pair-wise loss methods (Hsu et al., 2018; Han et al., 2019; Zhao & Han, 2021; Cao et al., 2022): These methods explore various techniques, such as robust ranking statistics (Han et al., 2019) and cosine similarity (Cao et al., 2022), to measure the similarity between two data points in the representation space, and minimize the distance of similar data point. 2) Self-labeling loss methods (Fini et al., 2021; Gu et al., 2023; Zhang et al., 2023a; Xu et al., 2024): These methods formulate the problem of generating balanced or imbalanced pseudo labels as an optimal transport problem and learn from these pseudo labels. 3) Self-distillation loss methods (Wen et al., 2022; Zhang et al., 2022a): These methods generate both sharp and soft predictions for two augmented views of the same data. The sharp prediction, which is typically more definitive and confident, is then used to supervise the soft prediction. In addition to the above approaches, other strategies have been proposed for learning representations of novel classes. For example, (Vaze et al., 2022; Pu et al., 2023) introduce various contrastive learning strategies and perform clustering using semi-kmeans. However, these methods primarily focus on static scenarios and have limitations when applied to real-world applications where data is collected in a streaming manner.

**Continual Category Discovery** Continual category discovery (CCD) aims to discover novel classes in a continual manner. (Joseph et al., 2022; Roy et al., 2022) first proposed the CCD setting, framed in two sessions: the first with supervision and the second involving fully unlabeled new classes. GM (Zhang et al., 2022b) proposed a more general setting, assuming the incremental stages have unlabeled data containing both known and new classes. Then, Zhao & Mac Aodha (2023); Marczak et al. (2023); Cendra et al. (2024) generalized this problem by proposing a setting where all tasks contain both labeled and unlabeled data. Subsequently, Liu et al. (2023) leveraged pretrained models and learned a classifier using self-labeling loss to discover novel classes.

Although Liu et al. (2023) shares similarities with our method, it falls short in utilizing advanced techniques from continual learning and novel class discovery to effectively address RF-CCD, resulting in a less convincing comparison. Additionally, their reliance on self-labeling loss to cluster novel classes enforces a strong equality constraint on cluster size, which proves ineffective due to noisy learning (Appendix E). More critically, they only fix the backbone without providing any analysis or insights into the role of representation learning for RF-CCD. As a result, their work offers limited insights for future research, which are key contributions of our work. Moreover, our method outperforms Liu et al. (2023) with a simpler design.

Table 1: Baseline results on CUB and Scars. The datasets are divided into four equally sized sessions (refer to Sec.5.1 and Appendix D for more details). All experiments are conducted with DINO (Caron et al., 2021) pretrained model.

| Method | CUB200 | | | Scars196 | | |
|---|---|---|---|---|---|---|
| | Last Acc | Old | New | Last Acc | Old | New |
| Lwf (Li & Hoiem, 2017) + PwL | 34.6 | 60.1 | 26.5 | 18.1 | 49.0 | 7.8 |
| Lwf (Li & Hoiem, 2017) + SeLa | 26.0 | **86.4** | 6.9 | 21.2 | **79.5** | 1.8 |
| Lwf (Li & Hoiem, 2017) + SeDist | 38.4 | 69.4 | 28.6 | 21.2 | 55.0 | 9.9 |
| CODA-P (Smith et al., 2023) + PwL | 42.9 | 72.9 | 33.2 | 10.2 | 18.5 | 7.4 |
| CODA-P (Smith et al., 2023) + SeLa | 34.8 | 83.2 | 19.1 | 18.3 | 57.0 | 5.3 |
| CODA-P (Smith et al., 2023) + SeDist | 40.3 | 43.3 | 31.1 | 14.8 | 13.5 | 15.2 |
| SLCA (Zhang et al., 2023b) + PwL | 48.0 | 70.6 | 40.6 | 21.5 | 39.0 | 15.6 |
| SLCA (Zhang et al., 2023b) + SeLa | 50.4 | 76.0 | 42.1 | 26.3 | 59.4 | 15.1 |
| SLCA (Zhang et al., 2023b) + SeDist | **55.5** | 75.3 | **49.1** | **31.3** | 64.1 | **20.2** |
| MetaGCD (Wu et al., 2023) | 42.9 | 48.6 | 40.6 | 13.5 | 16.1 | 12.5 |
| Frost (Roy et al., 2022) | 50.2 | 75.0 | 42.1 | 20.9 | 43.0 | 13.4 |
| KTRFR (Liu et al., 2023) | 44.2 | 72.8 | 34.5 | 25.9 | 59.2 | 14.6 |

## 3 UNRAVELING THE CHALLENGES OF RF-CCD

In this section, we begin by integrating advanced methods from two domains, offering a convincing experimental comparison with existing approaches. Following this, we perform additional experiments to assess representation quality, emphasizing the unique challenges presented by RF-CCD.

### 3.1 PROBLEM FORMULATION

In RF-CCD, to leverage the development of foundation models, we start with a self-supervised pre-trained model $g_\theta$ (Caron et al., 2021; Zhou et al., 2021; He et al., 2022). The model is initially given a labelled dataset $\mathcal{D}_0 = \{x_i^0, y_i^0\}_{i=1}^{N_0}$ for supervised learning on session $t = 0$, where $x_i^s$ is the input image and $y_i^0$ is the label within $\mathcal{Y}_0$. After $t = 0$ is finished, the labelled set is discarded and model is presented with a sequential of $(T - 1)$ NCD sessions, each of which contains an unlabelled dataset $\mathcal{D}_t = \{x_i^t\}_{i=1}^{N_t}$. For different sessions $i, j$, we assume classes are disjoint, $i.e.$, $\mathcal{Y}_i \cap \mathcal{Y}_j = \emptyset$. During each session $t$, it is not allowed to store data from previous sessions. The aim of RF-CCD is to continuously discover novel classes in $\mathcal{D}_t$, without compromising performance on previously seen classes from $\mathcal{D}_0$ to $\mathcal{D}_{t-1}$.

### 3.2 MODIFY CL METHODS TO HANDLE RF-CCD

**Combination of CL and NCD methods** RF-CCD is a combination of continual learning and novel class discovery. To delve deeper into the challenges of RF-CCD, we integrate various continual learning methods with different NCD techniques to establish more comprehensive methods. Specifically, we select several typical rehearsal-free continual learning approaches, including well-known Learning without Forgetting (LwF) (Li & Hoiem, 2017), as well as two of the latest approaches: CODA-prompt (Smith et al., 2023) and SLCA (Zhang et al., 2023b). As much of the subsequent analysis is based on SLCA, we provide a brief overview of it in Appendix B.

As outlined in Section 2.2, NCD techniques can be broadly categorized based on the loss functions used for clustering novel classes. Specifically, we categorize the existing loss functions into three groups: 1) self-labeling loss (*SeLa*), 2) pairwise loss (*PwL*), and 3) self-distillation loss (*SeDist*). These loss functions have been summarized in Sec. 2.2 and detailed in Appendix A. To enable standard continual learning methods to effectively cluster novel classes, we substitute the conventional cross-entropy loss with the aforementioned three unsupervised losses, respectively.

**State of the Arts RF-CCD methods** There is existing research in the field of Continual Category Discovery, including methods such as Frost (Roy et al., 2022), GM (Zhang et al., 2022b), iGCD (Zhao & Mac Aodha, 2023), MetaGCD (Wu et al., 2023), and KTRFR (Liu et al., 2023). We exclude the comparison with GM and iGCD, as their methods heavily rely on memory buffers, making them difficult to handle RF-CCD.

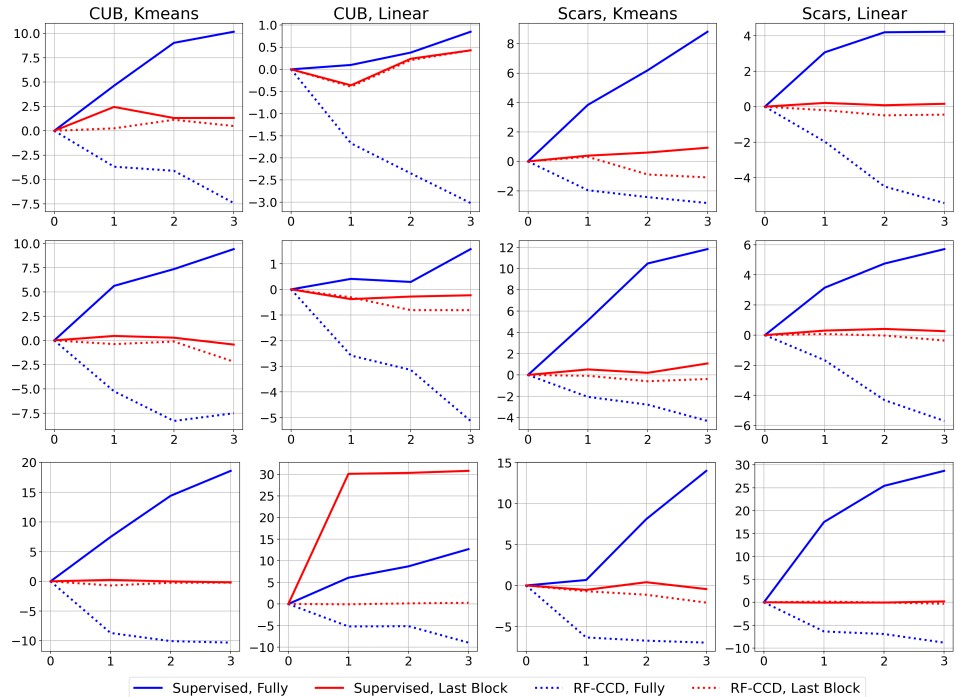

Figure 1: Representation Analysis: We analyze representation using DINO (row 1), iBOT (row 2), and MAE (row 3) pre-trained backbones with K-means and Linear Probing on the CUB and Scars datasets. The x-axis indicates the stages of continual learning, while the y-axis shows the accuracy difference between the current stage and the initial stage. "Supervised" and "RF-CCD" denote supervised and rehearsal-free continual category discovery settings, respectively. "Fully" and "Last Block" refer to finetuning the entire network or just the last block.

**Results Analysis** We conduct extensive experiments by using the DINO model (Caron et al., 2021), which is pretrained on ImageNet1K in an unsupervised manner. As shown in Table 1, while methods in the RF-CCD field, such as Frost (Roy et al., 2022) and KTRFR (Liu et al., 2023), have achieved commendable results, they are significantly outperformed by the SLCA (Zhang et al., 2023b) with self-distillation losses. Specifically, compared to Frost (Roy et al., 2022), SLCA+SeDist shows an improvement of 7.0% on CUB200 and 6.8% on Scars196 for novel classes.

In addition, we found that the optimal choice of unsupervised loss is closely related to the continual learning framework and dataset. Among them, self-distillation loss (Caron et al., 2021), which demonstrates superior performance within the SLCA (Zhang et al., 2023b) and LwF (Li & Hoiem, 2017) frameworks, emerges as a strong candidate for subsequent analysis.

### 3.3 UNRAVEL THE CHALLENGE OF RF-CCD FROM REPRESENTATION PERSPECTIVE

**Motivation** In continual learning, with an appropriate learning strategy, representations can be progressively enhanced over time in the presence of labeled data streams (Rebuffi et al., 2017; Zhu et al., 2021). However, in RF-CCD, it remains unclear whether representation improves within the current learning framework due to the noise involved in discovering novel classes. To shed light on this issue, building on the experiments in Sec.3.2, we further investigate the optimal baseline, SLCA (Zhang et al., 2023b) combined with Self-distillation (Caron et al., 2021), and analyze how representation quality evolves with unlabeled data.

Specifically, after each incremental task, we utilize K-means and linear probing to evaluate representation quality. For a comprehensive analysis, we compare the representations of two learning paradigms: (1) Supervised continual learning (original SLCA), which serves as the upper bound, and (2) SLCA + SeDist, which acts as a strong approach for the RF-CCD task. Additionally, we conduct experiments using various unsupervised pre-trained models, including DINO (Caron et al.,

Figure 2: FAC framework. In the first stage, we fine-tune a ViT on labeled known categories. In the subsequent stages, we perform k-means clustering on new categories and then derive Gaussian means and variances for each cluster. Finally, we sample data from each Gaussian (including the current category as well as from memory) to train the classifier.

2021), iBOT (Zhou et al., 2021), and MAE (He et al., 2022), and apply two fine-tuning strategies: full fine-tuning and fine-tuning of only the last block.

**Results and Analysis** The performance of linear probing and K-means is illustrated in Fig. 1. Overall, experiments with the three backbones and two evaluation methods exhibit similar trends, leading us to several conclusions. Specifically, in supervised learning, when the model is fully fine-tuned, the representation quality gradually improves with the addition of incremental data. However, when only the last block is fine-tuned, such improvements are marginal. In contrast, observations differ in the RF-CCD setting. Here, fully fine-tuning results in a significant degradation of representation quality, while fine-tuning only the last block yields no noticeable improvement and may even lead to a decline in representation quality. We believe that continuous learning with unlabeled data accumulates noise, which is detrimental to representation quality. Moreover, the more parameters that are tuned, the more harmful this noise becomes.

If we aim to continuously improve the representation ability in RF-CCD, as supported by supervised results, we must adjust more parameters to increase the upper limit. However, this adjustment leads to poor outcomes with the best existing strategies, making improvement particularly challenging.

In conclusion, this analysis underscores a key challenge in RF-CCD: *how can we continuously improve or maintain the representation ability of the RF-CCD model?*

## 4 METHOD

In this section, we introduce our framework for Rehearsal-Free Continual Category Discovery (RF-CCD), which achieves strong performance with a simple design. As shown in Fig.2, we fine-tune the model on labeled data in the initial session. In later sessions, we freeze the backbone and use k-means clustering to generate pseudo labels from the representation space. Then, we assume each cluster follows a Gaussian distribution and derive the mean and variance for each cluster. Finally, we sample data from the current and stored Gaussian distributions to train the classifier. Additionally, we propose a novel method to estimate the number of novel classes in RF-CCD scenarios.

### 4.1 FREEZE AND CLUSTER

**Freeze Representation** As illustrated in Sec.3.3, the representation ability shows no improvement and even degradation in CCD after the first session. Therefore, we simply learn the representation in the supervised session ($t = 0$) and freeze the backbone $g_\theta$ for all remaining tasks. Although freezing the backbone sacrifices the model's plasticity, it avoids the detrimental effects caused by noisy novel class learning and forgetting simultaneously, thereby enhancing stability.

**Pseudo Label Generation by Clustering and Classifier Learning** Since there are no labels for novel class data, and thanks to the powerful representation, we first generate pseudo-labels for the novel classes through k-means clustering in the representation space. After obtaining the representation, rather than directly using these pseudo-labels for classifier learning, we follow the approach in (Zhang et al., 2023b) to train the classifier.

---

**Algorithm 1** Class Number Estimation Algorithm

---

**Input:** Dataset $\mathcal{D}$, initial cluster number $m$, merging threshold $d_{min}$
**Output:** Estimated number of novel classes $K_u$
Over-cluster the data to obtain sub-clusters $C = \{c_1, c_2, \ldots, c_m\}$
**while** true **do**
    Compute the distance matrix $\mathcal{D}$ between sub-clusters
    Find the closest pair of sub-clusters $(c_i, c_j)$
    **if** $\mathcal{D}(c_i, c_j) > d_{min}$ **then**
        **break**
    **else**
        Merge $c_i$ and $c_j$ into a single sub-cluster
    **end if**
**end while**
**return** the number of remaining sub-clusters as $K_u$

---

Specifically, we model each cluster distribution as a single Gaussian, estimating the mean and variance for each cluster. We then sample data from both the current learning stage and past learning stages using the stored mean and variance to train the classifier. This approach helps mitigate the forgetting of the classifier. Additionally, we apply logit normalization (Wei et al., 2022) to prevent bias towards known classes. As the classifier learning is not our contribution, we detail it in Appendix C.

Despite the simplicity of our baseline, in experiments (Sec.5.2), we show that we achieve impressive results compared to advanced methods from both domains. Meanwhile, we emphasize that our contribution not mainly lies in the baseline itself but in our extensive analysis, which illustrates the challenges of CCD and provides a convincing baseline for future work.

### 4.2 NOVEL CLASS NUMBER ESTIMATION

In RF-CCD, it is usually assumed that the number of novel classes in each task is known. However, in the real world, this assumption does not always hold. Therefore, we propose a novel method to estimate the number of novel classes in RF-CCD.

Specifically, for known classes data, we first perform over-clustering to obtain many clusters, which is generally set to three times the true number, i.e., $3 \times C^t$. Then, we calculate the Euclidean distance between cluster centers and greedily merge the two clusters with the minimal distance. The merging process is stopped until the number of clusters is equal to the ground truth. Therefore, we obtain the minimal distance $d_{\min}$ between the clusters.

We assume that if the distance between two clusters is smaller than $d_{\min}$, the two clusters are from the same class with high probability. Based on this assumption, in subsequent tasks, we first perform over-clustering to obtain multiple sub-clusters and continuously merge the two closest sub-clusters until the distance between the closest two clusters exceeds the merging threshold $d_{\min}$. The number of novel classes is determined by the number of clusters remaining after the merging process. The detail of algorithm is shown in Algo.1.

## 5 EXPERIMENTS

### 5.1 EXPERIMENT SETUP

**Dataset** We build baselines and validate the effectiveness of our method on three fine-grained datasets: CUB200 (Welinder et al., 2010), Stanford Cars196 (Krause et al., 2013), and iNat550, as well as one generic dataset: Tiny-ImageNet200 (Le & Yang, 2015). We construct the iNat550 dataset by sampling 50 subcategories from each of the 11 supercategories in iNaturalist21 (Van Horn et al., 2021). We divide CUB and Stanford Cars196 into four equal sessions. To reflect more realistic scenarios, we adopt a ten-session strategy to generate the task sequence for Tiny-ImageNet200. For iNat550, we create an 11-session task, with each session exclusively representing one supercategory. This setup reduces the semantic relationships across sessions and increases the difficulty of knowledge transfer.

Table 2: Main experimental results. The experiments are conducted across four datasets: CUB200 and Scars196 (4 sessions), and iNat550 and TinyImageNet200 (10 sessions). The first group of results represents the supervised upper bound. The second group combines continual learning with novel class discovery methods. The third group includes original CCD methods, while the fourth group focuses on classifier learning methods.

| Method | CUB200 | | | Scars196 | | | iNat550 | | | Tiny-ImageNet 200 | | |
|---|---|---|---|---|---|---|---|---|---|---|---|---|
| | Last | Old | New | Last | Old | New | Last | Old | New | Last | Old | New |
| SLCA (Zhang et al., 2023b) | 80.9 | - | - | 77.7 | - | - | 70.1 | - | - | 79.1 | - | - |
| RanPAC (McDonnell et al., 2024) | 80.9 | - | - | 40.4 | - | - | 70.6 | - | - | 81.8 | - | - |
| LwF (Li & Hoiem, 2017) + SeDist | 38.4 | 69.4 | 28.6 | 21.2 | 55.0 | 9.9 | 8.9 | 2.2 | 9.6 | 31.2 | 68.1 | 27.1 |
| CODA-P (Smith et al., 2023) + SeLa | 34.8 | 83.2 | 19.1 | 18.3 | 57.0 | 5.3 | 19.6 | 53.2 | 16.2 | 23.7 | 82.5 | 17.2 |
| CODA-P (Smith et al., 2023) + PwL | 42.9 | 72.9 | 33.2 | 10.2 | 18.5 | 7.4 | 30.4 | 63.2 | 27.1 | 12.6 | 4.6 | 13.5 |
| CODA-P (Smith et al., 2023) + SeDist | 40.3 | 43.3 | 31.1 | 14.8 | 13.5 | 15.2 | 24.4 | 25.8 | 24.3 | 57.4 | 47.7 | 58.8 |
| SLCA (Zhang et al., 2023b) +SeLa | 50.4 | 76.0 | 42.1 | 26.3 | 59.4 | 15.1 | 26.6 | 63.6 | 22.9 | 33.3 | 33.3 | 33.3 |
| SLCA (Zhang et al., 2023b) + PwL | 48.0 | 70.6 | 40.6 | 21.5 | 39.0 | 15.6 | 30.1 | 56.2 | 27.3 | 34.2 | 23.1 | 35.4 |
| SLCA (Zhang et al., 2023b) + SeDist | 55.5 | 75.3 | 49.1 | 31.3 | 64.1 | 20.2 | 34.4 | 66.4 | 31.1 | 50.2 | 49.6 | 50.3 |
| MetaGCD (Wu et al., 2023) | 42.9 | 48.6 | 40.6 | 13.5 | 16.1 | 12.5 | - | - | - | - | - | - |
| Frost (Roy et al., 2022) | 50.2 | 75.0 | 42.1 | 20.9 | 43.0 | 13.4 | 31.7 | 54.2 | 29.5 | 58.9 | 49.9 | 59.9 |
| KTRFR (Liu et al., 2023) | 44.2 | 72.8 | 34.5 | 25.9 | 59.2 | 14.6 | 26.5 | 70.0 | 22.1 | 46.0 | 73.8 | 42.9 |
| NCM (Janson et al., 2022) | 57.6 | 78.1 | 50.9 | 29.5 | 62.7 | 18.3 | 37.3 | 70.4 | 34.0 | 68.1 | 74.8 | 67.4 |
| FeCAM (Goswami et al., 2024) | 53.8 | 75.0 | 46.9 | 29.2 | 63.4 | 17.6 | 36.6 | 69.4 | 33.3 | 69.1 | 76.6 | 68.3 |
| RanPAC (McDonnell et al., 2024) | 62.8 | **81.8** | 56.6 | 34.2 | **78.0** | 19.3 | 35.6 | **75.4** | 31.6 | 72.8 | 77.2 | 72.3 |
| FAC (Ours) | **66.2** | 81.2 | **59.6** | **35.6** | 73.7 | **22.7** | **39.5** | 72.6 | **36.2** | **73.7** | 77.5 | **73.2** |

In all considered splits, classes are evenly distributed, and the first session is considered supervised. We show the details of dataset in Appendix D.

**Evaluation Metric** Following common settings in Continual Learning (Panos et al., 2023), we report *Last Acc*, the Top-1 accuracy of the final model on a joint test set containing all categories. During inference, we follow the task-agnostic protocol, i.e., the task ID is unknown in the joint test set. To measure open-world recognition ability and distinguish between labeled and unlabeled classes, we further report the prediction accuracy for both the 'Old' subset (instances belonging to the supervised session) and the 'Novel' subset (samples from all unsupervised stages).

The mapping from unsupervised clustering ID to ground truth ID is done via the Hungarian optimal assignment algorithm (Kuhn, 2010) after learning from each unsupervised session. This mapping for unsupervised data is preserved after each session and used for inference in subsequent sessions.

**Implementation Details** For methods like CODA-prompt (Smith et al., 2023) and SLCA (Zhang et al., 2023b), we followed all original training settings, only replacing the supervised training signal with an unsupervised learning loss. For the state-of-the-art CNCD method Frost (Roy et al., 2022), we inherited most of its training hyperparameters, except for searching for the best learning rate. For NCM (Janson et al., 2022), FeCAM (Goswami et al., 2024), and RanPAC (McDonnell et al., 2024), which only learn classifiers, we first generate pseudo-labels using k-means and then follow their methods to train the classifier. All methods are trained with a ViT-B-16 backbone using DINO (Caron et al., 2021) pre-trained weights. For methods that require tuning backbone, we only fine-tune the last block for a fair comparison.

For our proposed baseline (FAC), during supervised adaptation, we fine-tune the last transformer block. In the subsequent unsupervised data stream, we adopt the SGD optimizer and use a cosine decay learning scheduler with an initial learning rate of 0.1 for classifier learning. We set the logit normalization temperature $\tau$ to 0.1 in all experiments.

## 5.2 COMPARE WITH THE STATE OF THE ARTS AND STRONG BASELINES

As shown in Tab.2, we have conducted extensive comparative experiments with various methods, and the excellent results prove the effectiveness of our baseline. Compared to the supervised upper bound, our method achieves satisfactory results in CUB200 and Tiny-ImageNet200, while the results on Stanford Cars196 and iNat550 still fall short of the upper bound. Additionally, when compared to the best combined method (SLCA + SeDist), we outperform them by 10.7, 4.3, 5.1, and 23.5 on CUB, Stanford Cars196, iNat550, and Tiny-ImageNet200, respectively.

Table 3: Experiments with the estimated number of novel class.

|  | CUB200 | Scars196 | iNat550 | Tiny-ImageNet200 |
|---|---|---|---|---|
| GT Class Number | 50 | 49 | 50 | 20 |
| Average Estimated Number | 68 | 70 | 70 | 21 |
| Known Class Number Last Acc | 66.2 | 35.6 | 39.5 | 73.7 |
| Unknown Class Number Last Acc | 62.4 | 33.8 | 36.9 | 67.0 |

Table 4: Ablation study. Here we present the Last-Acc after continual learning of all sessions. 'SA' represents supervised session adaptation, 'LN' is logit normalization, and 'GR' stands for generative replay, without 'GR' is simply train with pseudo label of training set on each session.

| SA | GR | LN | CUB200 | | | Scars196 | | |
|---|---|---|---|---|---|---|---|---|
|  |  |  | Last | Old | New | Last | Old | New |
| ✓ |  |  | 33.0 | 0.0 | 44.5 | 12.1 | 0.0 | 16.2 |
|  | ✓ |  | 51.6 | 77.3 | 42.6 | 22.8 | 61.3 | 9.8 |
| ✓ | ✓ |  | 59.4 | 72.4 | 54.9 | 31.7 | 61.5 | 21.6 |
| ✓ | ✓ | ✓ | **66.2** | **81.2** | **59.6** | **35.6** | **73.7** | **22.7** |

We also compare our approach with native RF-CCD methods. Notably, KTPFR (Liu et al., 2023) is similar to our method but employs the SeLa loss (Fini et al., 2021) to generate pseudo-labels through classifier learning. However, our approach significantly outperforms theirs. As shown in Appendix E, the simple K-means algorithm is more effective at producing high-quality pseudo-labels than the SeLa loss-based classifier, which may be adversely affected by the noisy learning of unlabeled data.

Furthermore, compared to the classifier learning methods in the fourth group, we achieve substantial improvements in both final and novel class accuracy. The results demonstrate that, unlike FeCAM (Goswami et al., 2024), which utilizes Mahalanobis distance for classifier learning, or RanPAC (McDonnell et al., 2024), which projects features into a high-dimensional space, our approach—simply normalizing the features and learning the classifier in the normalized feature space—yields better results.

### 5.3 CLASS NUMBER ESTIMATION

The above experiments assume that the number of novel classes is known, which is not realistic in practice. To adapt a model to an open-world environment, in each stage, we estimate the number of novel classes and utilize the estimated number for clustering these novel classes. We show the average of the estimated number of novel classes and the final accuracy. The results are presented in Table 3. Although the average estimated number is larger than the ground truth (GT), the final accuracy is comparable to the setting where the GT is known. As our method estimates many clusters, there are numerous small subclusters for each cluster. In the Hungarian matching, these small subclusters are ignored, resulting in minimal impact on the final accuracy.

### 5.4 ABLATION STUDY

We conduct an ablation study to demonstrate the effectiveness of supervised adaptation (SA), generative replay (GR), and logit normalization (LN), as presented in Table 4. Comparing rows 1 and 3, we observe that GR significantly improves performance on both old and new classes, effectively mitigating catastrophic forgetting, particularly for known classes. Comparing rows 2 and 3, SA notably enhances novel class performance due to better representation initialization, though it reduces performance on known classes in CUB200, likely due to classifier bias towards novel classes. With LN, this bias is largely alleviated, resulting in significant improvements in old class performance. The ablation study highlights the contribution of each component in our baseline, demonstrating the benefits of supervised adaptation, generative replay, and logit normalization.

## 6 CONCLUSION

In this work, we leverage advanced techniques from continual learning and novel class discovery, conducting extensive experiments to tackle the rehearsal-free continual category discovery (RF-CCD) problem. Our experiments illustrate that: 1) migrating the SLCA (Zhang et al., 2023b) method and using self-distillation loss (Wen et al., 2022) to learn new classes can surpass the previous RF-CCD method; 2) *"in the presence of strong foundation models and with current novel class learning strategy, the representation is hard to improve and even degrades in continuous discovery."* Therefore, we propose a simple baseline, named Freeze and Cluster (FAC), to tackle RF-CCD. This approach estimates the number of novel classes, learns representations in the initial stage, and then only learns the classifier using cluster labels in subsequent stages. Despite its simplicity, it outperforms all existing methods. We hope our detailed experimental analysis and strong baseline can motivate future work to develop more effective methods to tackle this problem. Meanwhile, since the representation quality is difficult to improve, we also hope to re-examine the learning paradigm of RF-CCD and consider to incorporate a limited amount of human supervision signals (Ma et al., 2024) to achieve more effective open-world learning.

**Limitation** Although our analysis is comprehensive and provides insights into the problem, our experimental analysis has some limitations: 1) Our characterization analysis primarily relies on the optimal combination strategy (SeLa + SeDist). While it helps explain the issue, it has not been experimentally verified across more combination methods to fully establish the universality of our conclusions. We acknowledge that reaching universal conclusions is challenging because we cannot exhaust all methods due to limited computational resources. 2) Our experimental analysis is based on a simplified setting, assuming that the unlabeled data consists solely of novel class data. We have not investigated a more generalized setting where the unlabeled data includes both novel and known classes. We believe that a generalized approach could first identify whether the data belongs to a novel or known class before adapting it to our experimental framework. While incorporating known-class data may help mitigate forgetting to some extent, it does not address the inherent challenges associated with noisy learning of novel class data, which is a crucial factor in degrading representation quality.

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

## A  SUMMARY OF LOSS FOR NOVEL CLASS LEARNING

In this section, we provide details on the learning loss for novel classes. We summarize the loss into three categories: pairwise similarity loss (Hsu et al., 2018; Han et al., 2021; Cao et al., 2022), which minimize the distance of a pair of similar data, a self-labeling loss (Asano et al., 2019; Fini et al., 2021), which utilizes Sinkhorn-knopp algorithm to generate pseudo label for unlabeled data, and self-distillation loss. Before detailing these losses, we introduce some notations: $x^u$ represents an unlabeled image, $y^u$ represents the model's prediction, $z^u$ represent the corresponding representation, and $f$ denotes the feature extractor.

**Pairwise similarity loss (Hsu et al., 2018):**  The pairwise similarity loss learns to group a pair of similar data, thus learning compact representation for unlabeled data. Specifically, given a batch of $B$ unlabeled data, we forward the model to get the embedding $z^u = f(x^u)$ and prediction $\mathbf{y}^u = p(y^u; x^u)$. For each unlabeled data, to get its the pairwise pseudo label, we find its nearest neighbor in the embedding space from the $B$ unlabeled data. We denote the nearest neighbor of $z_i^u$ as $\hat{z}_i^u$. Therefore, ignoring the negative pairs (Cao et al., 2022), the formulation of pairwise loss is:

$$\mathcal{L}_u = \frac{1}{|\mathcal{D}^u|} \sum_{i=0}^{|\mathcal{D}^u|} -\log(\mathbf{y}_i^u)^T \hat{\mathbf{y}}_i^u \tag{1}$$

To avoid all the unseen class degenerate to a single cluster, Cao et al. (2022) also introduce a simple entropy regularization term to regularize the size of cluster.

**Self-labeling loss (Asano et al., 2019):**  The self-labeling loss first generates pseudo-labels for unlabeled data, then utilizes the generated pseudo label to self-train model. It assumes unlabeled data are equally partitioned into each cluster and utilizes Sinkhorn-knopp algorithm to find an approximate assignment. We denote $\mathbf{y}^q = q(y^u; x^u), \mathbf{y}^p = p(y^u; x^u)$, and $\mathbf{y}^p, \mathbf{y}^q \in \mathbb{R}^{(m+n)\times 1}$. Let $\mathbf{Q} = [\mathbf{y}_1^q, \mathbf{y}_2^q, ,, \mathbf{y}_B^q] \frac{1}{B}$, $\mathbf{P} = [\mathbf{y}_1^p, \mathbf{y}_2^p, ,, \mathbf{y}_B^p] \frac{1}{B}$ be the joint distribution of $B$ sampled data. We estimate $\mathbf{Q}$ by solving an optimal transport problem. We refer readers to (Cuturi, 2013; Asano et al., 2019) for details. The optimal $\mathbf{Q}$ is the pseudo label of unlabeled data. We denote the optimal pseudo label as $q^*(y^u; x^u)$, so the self-labeling is formulated as:

$$\mathcal{L}_u = \frac{1}{|\mathcal{D}^u|} \sum_{i=0}^{|\mathcal{D}^u|} -q^*(y_i^u; x_i^u) \log p(y_i^u; x_i^u) \tag{2}$$

**Self-distillation**  For each unlabeled data point $x_i$, we generate two views $x_i^{v_1}$ and $x_i^{v_2}$ through random data augmentation. These views are then fed into the ViT (Dosovitskiy et al., 2020) encoder and cosine classifier ($h$), resulting in two predictions $y_i^{v_1} = h(f_\theta(x_i^{v_1}))$ and $y_i^{v_2} = h(f_\theta(x_i^{v_2}))$, $\mathbf{y}_i^{v_1}, \mathbf{y}_i^{v_2} \in \mathbb{R}^{C^k + C^n}$. As we expect the model to produce consistent predictions for both views, we employ $\mathbf{y}_i^{v_2}$ to generate a pseudo label for supervising $\mathbf{y}_i^{v_1}$. The probability prediction and its pseudo label are denoted as:

$$\mathbf{p}_i^{v_1} = \texttt{Softmax}(y_i^{v_1}/\tau), \quad \mathbf{q}_i^{v_2} = \texttt{Softmax}(y_i^{v_2}/\tau') \tag{3}$$

Here, $\tau, \tau'$ represents the temperature coefficients that control the sharpness of the prediction and pseudo label, respectively. Similarly, we employ the generated pseudo-label $\mathbf{q}_i^{v_1}$, based on $\mathbf{y}_i^{v_1}$, to supervise $\mathbf{y}_i^{v_2}$. However, self-labeling approaches may result in a degenerate solution where all novel classes are clustered into a single class (Caron et al., 2018). To mitigate this issue, we introduce an additional constraint on cluster size. Thus, the loss function can be defined as follows:

$$\mathcal{L}_u = \frac{1}{2|\mathcal{D}^u|} \sum_{i=1}^{|\mathcal{D}^u|} [l(\mathbf{p}_i^{v_1}, \texttt{SG}(\mathbf{q}_i^{v_2})) + l(\mathbf{p}_i^{v_2}, \texttt{SG}(\mathbf{q}_i^{v_1}))] + \epsilon \mathbf{H}(\frac{1}{2|\mathcal{D}^u|} \sum_{i=1}^{|\mathcal{D}^u|} \mathbf{p}_i^{v_1} + \mathbf{p}_i^{v_2}) \tag{4}$$

Here, $l(\mathbf{p}, \mathbf{q}) = -\mathbf{q} \log \mathbf{p}$ represents the standard cross-entropy loss, and $\texttt{SG}$ denotes the "stop gradient" operation. The entropy regularizer $\mathbf{H}$ enforces cluster size to be uniform thus alleviating the degenerate solution issue. The parameter $\epsilon$ represents the weight of the regularize.

| Dataset | Labeled Session | | Unlabeled Session | |
|---|---|---|---|---|
| | #class | #image | #class | #image |
| CUB200 (Welinder et al., 2010) | 50 | 1.5k | 50 | 1.5k |
| StanfordCars (Krause et al., 2013) | 49 | 2.1k | 49 | 2k |
| Tiny-ImageNet (Le & Yang, 2015) | 20 | 10k | 20 | 10k |
| iNat550 (Van Horn et al., 2021) | 50 | 2.5k | 50 | 2.5k |

Table 5: Datasets used in our experiments. We provide the number of classes in the labeled and unlabeled sets.

## B  SLCA

SLCA (Zhang et al., 2023b) utilizes a two-stage learning process in continual learning tasks. In the first stage, representations are learned with a slow learning rate (e.g., 1e-4 for the SGD optimizer), and class means and variances are stored. These stored statistics are then replayed in the second stage for classifier learning, which helps mitigate forgetting and maintain the performance of both the backbone and the classifier. For further details, we refer readers to the original paper.

## C  CLASSIFIER LEARNING

In this section, we detail the classifier learning. Specifically, we sample generated features $\hat{F}_r = [\hat{f}_{t,1}, \ldots, \hat{f}_{t,M_c}]$ from the distribution $(\mu_c, \Sigma_c)$ of each cluster $c \in C_{1:T}$, where $M_c$ is the number of generated features per class. Note that $C_{1:T}$ represents all the observed clusters. These simulated features serve as input to adjust the classification layer $h_\theta$. The classifier training uses the common cross-entropy loss. Considering that the learned classes are repeatedly trained in each subsequent task, potentially leading to overconfidence in the training data, we follow the SLCA (Zhang et al., 2023b) and normalize the magnitude of the network output when computing the cross-entropy. $H_{1:T} = [l_1, \ldots, l_{[C_{1:T}]}]$ represents the logit scores of sampled features, which can be rewritten as the product of magnitude and direction: $H_{1:T} = \|H_{1:T}\| \cdot \vec{H}_{1:T}$. Here $\|H_{1:T}\| = \sqrt{\sum_{c \in C_{1:T}} \|l_c\|^2}$ represents the magnitude, and $\vec{H}_{1:T}$ represents the direction. We then perform classifier alignment using a modified cross-entropy loss with logit normalization:

$$\mathcal{L}(\theta_{cls}; \hat{F}_{1:T}) = -\log \frac{e^{l_y/(\tau\|H_{1:T}\|)}}{\sum_{c \in C_{1:T}} e^{l_c/(\tau\|H_{1:T}\|)}} \tag{5}$$

where $l_y$ denotes the $y$-th element of $H_{1:T}$ corresponding to label $y$. $\tau$ is the temperature hyperparameter.

## D  DATSET SPLITS

We provide the details of the dataset splits in Table 5 . For all the benchmarks considered, each session contains an equal number of classes.

## E  COMPARISON WITH KTRFR

In this section, we provide a detailed analysis of the pseudo-labeling effects of the KTR method compared to our simple K-means approach. The results indicate that the pseudo-label quality of K-means is superior. We speculate that the suboptimal performance of SeLA (Fini et al., 2021) arises from its reliance on optimal transport (OT) (Cuturi, 2013) to generate pseudo-labels and train a classifier with noisy pseudo-labels. The significant learning noise in this classifier degrades the quality of the pseudo-labels, leading to reduced effectiveness.

Table 6: Pseudo-Label quality on unlabelled session. Ours method uses clustering, while KTRFR (Liu et al., 2023) learns a linear classifier with Sela (Fini et al., 2021) Loss.

| Method | CUB200 | | | Scars196 | | |
|---|---|---|---|---|---|---|
| | stage1 | stage2 | stage3 | stage1 | stage2 | stage3 |
| KTRFR (Liu et al., 2023) | 41.7 | 46.0 | 45.0 | 20.3 | 22.4 | 24.1 |
| FAC (Ours) | 71.4 | 70.8 | 64.0 | 33.5 | 35.3 | 34.9 |

# F  COMPARISON WITH PROMPTCCD

We adapt PromptCCD (Cendra et al., 2024) to our setting and conduct comparative experiments. The results indicate that we achieve significant improvements over their approach. The results are presented in Table 7

Table 7: Comparison with PromptCCD (Cendra et al., 2024). We adapt the PromptCCD (Cendra et al., 2024) method to our benchmark and replace the Semi-supervised Kmeans with Kmeans to align with our evaluation protocol.

| Method | CUB200 | | | Scars196 | | | iNat550 | | |
|---|---|---|---|---|---|---|---|---|---|
| | Last | Old | New | Last | Old | New | Last | Old | New |
| PromptCCD (Cendra et al., 2024) | 40.5 | 48.4 | 37.9 | 12.2 | 15.2 | 11.2 | 31.0 | 41.0 | 30.0 |
| FAC (Ours) | **66.2** | **81.2** | **59.6** | **35.6** | **73.7** | **22.7** | **39.5** | **72.6** | **36.2** |