# OpenReview forum: "Freeze and Cluster: A simple baseline for Rehearsal-Free Continual Category Discovery"
_ICLR.cc/2025/Conference — ICLR 2025 Conference Withdrawn Submission_

### Official Review · Reviewer_KvAg · 2024-10-22

**Soundness:** 2
**Presentation:** 2
**Contribution:** 2
**Rating:** 3
**Confidence:** 5

**Summary:**

This paper tackles the problem of Rehearsal-Free Continual Category Discovery (RF-CCD). Firstly, extensive preliminary experiments are conducted to uncover the representation degradation in the unsupervised setting. To solve this issue, this paper proposes a simple yet effective method, namely, freeze and cluster. Specifically, the method freezes the feature space and performs progressive clustering together with cluster number estimation. Experimental results on various datasets showcase the superiority of the proposed FAC.

**Strengths:**

1. This paper is well-motivated and gives some in-depth analysis, e.g., the representation issue.
2. The preliminary experiments are comprehensive.
3. The method is targeted to solve the problem with remarkable performance.

**Weaknesses:**

1. Some conclusions are not convincing. For example, the paper contends that *We believe that continuous learning with unlabeled data accumulates noise, which is detrimental to representation quality.* The results might come from the limited exploration of combination methods. In rehearsal-free continual learning, feature-replay methods have shown great potential, like [R1] in continual learning and [R2] (FRoST) in CCD. A more recent work [R3] also employs feature replay to continually adjust the feature space, which also obtains remarkable performance for continual category discovery.
2. The proposed method is naïve and the novelty is relatively limited. The method includes basic clustering and number estimation. I’m afraid that this method could not provide so many insights to the community.
3. The feature space (i.e., backbone) is only tuned using labeled known classes, does this manner result in overfitting? because the data is purely labeled but the number is limited.
4. The class number estimation algorithm requires a pre-defined threshold $d_{min}$, which is intractable to define in advance and could largely impact the results. Some experiments and ablations should be included.
5. Detailed results of each continual session (at least for one or two datasets) should also be presented to show the performance.

References:
[R1]. Prototype Augmentation and Self-Supervision for Incremental Learning. CVPR 2021.
[R2]. Class-incremental Novel Class Discovery. ECCV 2022.
[R3]. Happy: A Debiased Learning Framework for Continual Generalized Category Discovery. NeurIPS 2024. arXiv:2410.0653.

**Questions:**

See Weaknesses.
1. The authors should explain how to define the threshold for class number estimation.
2. Overfitting issue of the supervised session t=0.
3. After clustering, why does it need to sample data from gaussian distribution then train the classifier? How about directly employ the clustering results in a classifier-free manner?

---

### Official Review · Reviewer_TDeM · 2024-10-23

**Soundness:** 3
**Presentation:** 2
**Contribution:** 2
**Rating:** 3
**Confidence:** 4

**Summary:**

This paper addresses the problem of Rehearsal-Free Continual Category Discovery (RF-CCD), where a model is required to learn sequentially from unlabeled data comprising novel classes, without retaining any previously seen data. The authors begin by exploring approaches from two distinct fields: continual learning and novel class discovery. They observe that a combination of methods from these fields can outperform the current state-of-the-art (SOTA) methods in RF-CCD. Building upon this combination, the authors leverage stronger self-supervised models and analyze the effect of updating the backbone using unlabeled data. Notably, they identify that fine-tuning the pre-trained backbone leads to performance degradation. To address this, the backbone is frozen to preserve its initial knowledge, and only the classifier is trained. Additionally, the proposed method employs an existing approach to model clusters of unlabeled data as Gaussian distributions, storing their means and variances. The classifier is trained by sampling data from these distributions. Furthermore, a class number estimation technique is introduced, which utilizes the minimum cluster distance among known classes to predict the number of new categories. The proposed method demonstrates superior performance on standard RF-CCD benchmarks.

**Strengths:**

1.	Exploring combinations of approaches from distinct research fields is an effective strategy for identifying potential improvements for the given task.
2.	Freezing the backbone during incremental sessions is a straightforward yet effective technique for maintaining model stability.
3.	The proposed method demonstrates superior performance across standard benchmarks.

**Weaknesses:**

1.	The interpretation of the observations in Fig. 1, as presented in L287-295, may not be entirely accurate. The primary objective of this experiment is to evaluate whether the unlabeled data can enhance the pre-trained representation. When trained with a supervised loss, the backbone aligns with the testing protocol by utilizing true labels, naturally converging to a state that reflects the task’s labels. In contrast, the unsupervised SeDist loss introduces discrepancies, as it neither aligns with the pre-training objective nor the evaluation classes, disrupting the pre-trained knowledge. Consequently, the learned representation is likely degraded due to the lack of constraints. To draw more concrete conclusions, it would be advisable to replace SeDist with the original pre-training loss function to maintain consistency between the two stages [A]. Additionally, it would be beneficial to conduct experiments in an incremental setting where both old and new classes are incorporated.
2.	The novelty of this paper is somewhat limited. While I acknowledge that the authors achieve better results compared to previous methods, as shown in Table 2, the proposed approach is relatively simple and intuitive. The main motivation of the paper centers on the observation that earlier methods do not leverage pre-trained models, and updating the backbone during incremental sessions degrades pre-learned knowledge. The solution proposed, utilizing a pre-trained model and freezing the backbone, is straightforward. In the field of continual learning, the challenge of balancing plasticity and stability has been extensively studied. Increasing the number of trainable parameters enhances plasticity but compromises stability, while reducing the number of trainable parameters has the opposite effect. This trade-off is well understood and has been widely explored in the literature. Freezing the backbone is a reasonable strategy to preserve pre-trained knowledge, but it limits the model's capacity to learn new information during incremental sessions. The authors are encouraged to explore more sophisticated balancing strategies, such as those used in methods like Grow and Merge, to better address the plasticity-stability trade-off.

Writing needs improvement:

1.	In L21-24 of the abstract, the claim regarding performance improvement through better integration of previous methods is not directly relevant to the conclusion that unlabeled data can harm the learned representation.
2.	In L58, the authors critique prior works for not addressing the unique challenges of RF-CCD. However, the paper does not explicitly define what these unique challenges are. It would be beneficial to reorganize the discussion, particularly with paragraph L60-71, to provide a clearer explanation.
3.	Additionally, the introduction does not sufficiently explain how unlabeled data can degrade the representation. It would be helpful to explicitly mention that updating the model using unlabeled data can disrupt the pre-trained backbone, as this critical detail is currently missing.
4.	It is recommended to establish a clearer connection between the conclusion in L300 and the methodology described in Section 4. For example, the writing should explain how the conclusion informs or motivates the proposed method.

[A] Finetune like you pretrain: Improved finetuning of zero-shot vision models. CVPR’23

**Questions:**

1.	In Table 1, do MetaGCD, Frost, and KTRFR also use the DINO model for a fair comparison?
2.	In Table 1, are the representations updated for the different combinations of continual learning and GCD methods?
3.	The class number estimation algorithm seems to rely on the base classes and assumes that the novel classes have the same level of granularity as the base classes. A potential concern is how the algorithm would perform if the novel classes are more fine-grained. Would it remain robust if the minimum distance is computed based on the base class distributions?

---

### Official Review · Reviewer_W8xN · 2024-10-27

**Soundness:** 3
**Presentation:** 3
**Contribution:** 2
**Rating:** 5
**Confidence:** 4

**Summary:**

In this paper,

**Strengths:**

1.The paper is easy to follow and the conclusions are easy to understand.

2.The experimental figures are clear and adequate in Fig.1.

**Weaknesses:**

1.It seems that the proposed method is a simple combination of existing continual learning and novel class discovery methods for RFCCD, which is somewhat not novel.

2.More related GCD works should be included for comparisons, i.e., [1-3]. And I wonder  what performance can be achieved when combining the best GCD and continual learning method for this task.

3.The datasets used for evaluation are somewhat small. The results on more large-scale datasets are more convincing (i.e., ImageNet-1K).

[1]Online Continuous Generalized Category Discovery

[2]PromptCCD: Learning Gaussian Mixture Prompt Pool for Continual Category Discovery

[3]Happy: A Debiased Learning Framework for Continual Generalized Category Discovery

There are some minor issues that will not affect the rating:
1."continuously identifying novel class by " should be "continuously identifying novel classes by ".

2."Specifically, for known classes data" should be "Specifically, for known-class data".

**Questions:**

1.What performance can be achieved when combining the best GCD and continual learning method for this task.

2.Why the performance of last block exhibits much smaller difference than "Fully" when trained in supervised and RF-CCD.

---

### Official Review · Reviewer_LKqT · 2024-11-04

**Soundness:** 3
**Presentation:** 3
**Contribution:** 2
**Rating:** 5
**Confidence:** 4

**Summary:**

The paper addresses the issue of continually identification of novel classes without using stored data, instead leveraging pre-trained models and focusing on representation quality. The authors argue that current approaches either train models from scratch or rely heavily on data storage, which in both cases is highly impractical for privacy-sensitive situations and does not effectively utilize advancements/strengths from foundational models. The authors provide a method named "Freeze and Cluster" for fine-tuning a pre-trained model on known classes, freezing the backbone, and using k-means clustering to generate pseudo-labels for novel classes. The classifier is then trained using these pseudo-labels while maintaining representational stability.

**Strengths:**

The problem under study in this work, Rehearsal-Free Continual Category Discovery, interestingly blends the challenges continual learning and novel class discovery problems. The authors’ work is original in that their ideas of freezing representations to avoid degradation when learning new classes and utilizing pseudo-labels from k-means clustering has been unexplored in this research area.

The authors' proposed method is in line with works in this field and showcases the efficacy of their approach. The experiment results are reasonable.

The writing is very clear.

The supporting text in the appendix also provides additional experimental details that more thoroughly analyzes the efficacy of their approach.

**Weaknesses:**

There are several technical concerns as listed below in "Questions".

A discussion on the computational efficiency of "Freeze and Cluster" is needed.

**Questions:**

1. If the number of classes is not known, how is dmin determined? Can the issue of finding the optimal number of class clusters be further developed?

2. "we model each cluster distribution as a single Gaussian", what if this does not hold? Please discuss potential limitations or alternative approaches if the Gaussian assumption doesn't hold. It is better to show how the method performs on datasets where class distributions are known to be non-Gaussian.

3. It is premature to draw a conclusion with one experiment: "degradation in CCD after the first session". When to decide to freeze is very important to the proposed method, which is not sufficiently discussed in the paper. Please provide a more detailed analysis of how different freezing strategies affect performance, and discuss criteria for determining the optimal point to freeze the model.

4. Exploration of other clustering methods instead of basic k-means could be interesting. Please show how different clustering algorithms might affect the performance of the method.

5. What if there is noise in pseudo-labels? One ablation study could be to compare the method's performance with and without artificially introduced noise in the pseudo-labels.

---

### Note · Authors · 2024-11-27

I have read and agree with the venue's withdrawal policy on behalf of myself and my co-authors.